# Processive dynamics of the usher assembly platform during uropathogenic *Escherichia coli* P pilus biogenesis

Minge Du [1,6], Zuanning Yuan [1,6], Glenn T. Werneburg [2,3,4,6], Nadine S. Henderson [2,3], Hemil Chauhan[2,3,5], Amanda Kovach [1], Gongpu Zhao [1], Jessica Johl[2,3], Huilin Li [1✉] & David G. Thanassi [2,3✉]

Uropathogenic *Escherichia coli* assemble surface structures termed pili or fimbriae to initiate infection of the urinary tract. P pili facilitate bacterial colonization of the kidney and pyelonephritis. P pili are assembled through the conserved chaperone-usher pathway. Much of the structural and functional understanding of the chaperone-usher pathway has been gained through investigations of type 1 pili, which promote binding to the bladder and cystitis. In contrast, the structural basis for P pilus biogenesis at the usher has remained elusive. This is in part due to the flexible and variable-length P pilus tip fiber, creating structural heterogeneity, and difficulties isolating stable P pilus assembly intermediates. Here, we circumvent these hindrances and determine cryo-electron microscopy structures of the activated PapC usher in the process of secreting two- and three-subunit P pilus assembly intermediates, revealing processive steps in P pilus biogenesis and capturing new conformational dynamics of the usher assembly machine.

[1] Department of Structural Biology, Van Andel Institute, Grand Rapids, MI, USA. [2] Department of Microbiology and Immunology, Stony Brook University, Stony Brook, New York, NY, USA. [3] Center for Infectious Diseases, Stony Brook University, Stony Brook, New York, NY, USA. [4] Present address: Department of Urology, Glickman Urological and Kidney Institute, Cleveland Clinic, Cleveland, OH, USA. [5] Present address: SUNY Downstate College of Medicine, Brooklyn, New York, NY, USA. [6] These authors contributed equally: Minge Du, Zuanning Yuan, Glenn T. Werneburg. ✉email: Huilin.Li@vai.org; David.Thanassi@stonybrook.edu

Bacterial pili, also known as fimbriae, are hair-like organelles that function in a range of critical activities, including the colonization of diverse environments, biofilm formation, and host-pathogen interactions. The conserved chaperone-usher (CU) pathway is responsible for the assembly of a large family of adhesive and virulence-associated pili in Gram-negative bacteria[1–3]. The P (Pap) and type 1 (Fim) pili of uropathogenic *Escherichia coli* (UPEC) are prototypical structures assembled by the CU pathway. P pili promote bacterial adherence to the globo-series of glycolipids in the kidney, leading to pyelonephritis, whereas type 1 pili mediate binding to mannosylated proteins in the bladder, leading to cystitis[4,5]. Given their central roles in initiating and sustaining infection, there is intense interest in understanding the molecular mechanisms of pilus assembly and function. Such knowledge may facilitate the development of therapeutics that disrupt pilus biogenesis as an alternative or complement to traditional antibiotics[6,7].

In the CU pathway, a dedicated chaperone facilitates the folding of pilus subunits in the periplasm and an integral outer membrane (OM) protein termed the usher provides the assembly platform and secretion channel for the pilus fiber. Newly synthesized pilus subunits enter the periplasm from the cytoplasm via the Sec translocon. The subunits undergo disulfide bond formation and then form binary complexes with the periplasmic chaperone[8] (PapD for P pili, FimC for type 1 pili) (Fig. 1a, b). The chaperone mediates subunit folding by a mechanism termed donor strand complementation (DSC), in which the chaperone donates a ß-strand to complete the immunoglobulin (Ig)-like fold of the subunit in a noncanonical manner[9,10] (Supplementary Fig. 1a). Periplasmic chaperone–subunit complexes are then recruited to the OM usher (PapC for P pili, FimD for type 1 pili) for subunit assembly into pili and secretion of the pilus fiber to the cell surface (Fig. 1b). The usher is a membrane-spanning, multidomain nanomachine that catalyzes the ordered exchange of chaperone–subunit for subunit-subunit interactions through a process termed donor strand exchange (DSE)[11–13]. DSE involves a strand invasion event in which the N-terminal extension (NTE) of an incoming pilus subunit displaces the donated chaperone ß-strand in the preceding subunit, linking the two subunits together via formation of a canonical Ig fold[14] (Supplementary Fig. 1a). ATP is not directly available at the bacterial OM and pilus biogenesis is driven by protein-protein interactions and the energetically favorable exchange of DSC for DSE mediated by the usher.

Ushers contain five domains: a central 24-stranded ß-barrel domain that inserts into the OM to form the secretion channel, an internal plug domain that forms the channel gate, a periplasmic N-terminal domain (NTD) that functions in subunit recruitment, and two tandem periplasmic C-terminal domains (CTD1 and CTD2) that function in pilus assembly and secretion[15–18]. In the inactive, apo usher, the plug domain is located within the ß-barrel pore, closing the channel, and the NTD resides free in the periplasm, ready to recruit chaperone–subunit complexes (Fig. 1b). The usher differentially recognizes chaperone–subunit complexes, with the adhesin recruited first, ensuring the assembly of functional pili with the adhesin at the tip[17,19]. In the type 1 pilus system, binding of a FimC-FimH (FimCH) chaperone–adhesin complex to the FimD usher activates the usher for pilus assembly[13,16,20]. Activation involves a handover event in which the chaperone–adhesin complex is transferred from the usher NTD to the CTDs, the plug domain is expelled from the channel to the periplasm, and the lectin domain of the adhesin inserts into the usher channel (Fig. 1b). Release of the NTD from the chaperone–adhesin complex bound at the CTDs resets the usher for recruitment of the next chaperone–subunit complex from the periplasm. The

newly recruited complex at the NTD and previously recruited complex at the CTDs are perfectly oriented by the usher to catalyze DSE, forming the first link in the pilus fiber[13,14]. The newly incorporated chaperone–subunit complex is then transferred from the NTD to the CTDs to reset the usher, together with the outward secretion of the pilus fiber through the usher channel. Iterative cycles of chaperone–subunit recruitment and DSE at the usher then allow for the ordered extension and secretion of the pilus fiber[21,22].

The stability and homogeneity of type 1 pilus assembly intermediates have led to structures of the initiating, one-subunit FimDCH usher–chaperone–adhesin complex and the three-subunit FimDCFGH usher–chaperone–tip complex[16,23,24]. In contrast, structural information for P pilus biogenesis at the usher is lacking. P pilus tip fibers, composed of the PapK, E and F subunits, and the PapG adhesin, are longer and more flexible compared to type 1 pili, and the variable incorporation of PapE subunits (~5–10 copies per pilus) creates structural heterogeneity (Supplementary Fig. 1b). P pili belong to a different clade in the CU superfamily and exhibit notable differences compared to type 1 pili, including that the PapG adhesin alone is not sufficient to activate the PapC usher and recruitment of a second subunit (PapF) may be necessary to activate the usher and form a stable assembly intermediate[3,25,26]. Only recently was a structure of the PapC usher in complex with the PapDG chaperone–adhesin complex solved, through in vitro mixing of purified PapC and PapDG components[18]. In contrast to the FimDCH usher–chaperone–adhesin complex from type 1 pili, the reconstituted P pilus complex was in a closed state, with the plug still lodged within the usher channel (Supplementary Fig. 8c, d). Thus, structures of in vivo P pilus assembly intermediates are needed to broaden understanding of CU pilus biogenesis and allow for the rational design of therapeutics that disrupt pilus biogenesis.

In this work, we took advantage of the modular nature of pilus biogenesis by the CU pathway to generate P pilus assembly intermediates suitable for structural analysis. Using cryo-electron microscopy (EM), we determined structures of the activated PapC usher in the process of secreting two- and three-subunit P pilus assembly intermediates. These structures show processive steps in P pilus biogenesis, reveal differences between P and type 1 pili, and capture new conformational states of the usher assembly machine.

## Results

**Generation of homogeneous and stable P pilus assembly intermediates**. We were unable to isolate stable PapCDG usher–chaperone–adhesin complexes from bacteria, supporting models in which the adhesin alone is insufficient to activate the PapC usher. Native usher–chaperone–P pilus tip assembly intermediates (PapCDKEFG) were stable to isolation; however, the native P pilus tip fibers are flexible and contain variable numbers of PapE subunits, making them unsuitable for structural analysis. Pilus biogenesis at the usher is a modular and iterative process[1,2]. The pilin domains of subunits are structurally similar but differences in their NTE donor strands and acceptor grooves determine their capacity to engage in DSE and assembly order in the pilus fiber[27]. We took advantage of this property to circumvent the inherent heterogeneity of the P pilus tip fiber and generate stable assembly intermediates suitable for structural analysis; i.e., lacking the variable PapE subunit. To generate a three-subunit intermediate, we replaced the PapK NTE with the NTE from PapE (PapK$_{E-NTE}$) and deleted *papE*, resulting in a ΔE P pilus tip fiber composed of PapK bound to PapF bound to PapG (Supplementary Fig. 1b). To generate a two-subunit intermediate, we replaced the PapK NTE with the NTE from

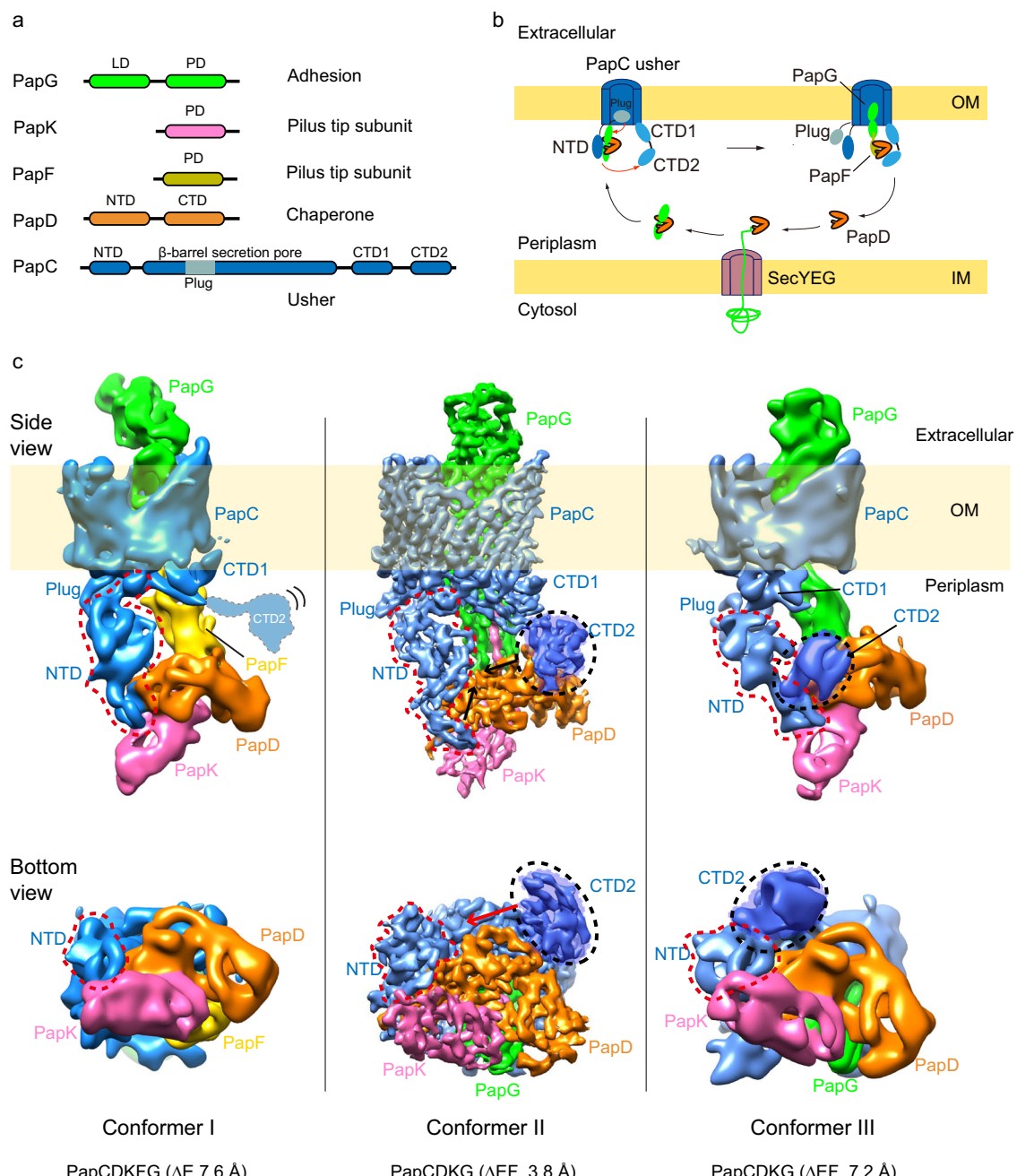

**Fig. 1 P pilus biogenesis and cryo-EM of the P pilus assembly intermediates. a** P pilus components. LD lectin domain, PD pilin domain, NTD N-terminal domain, CTD C-terminal domain, Plug, the plug domain embedded within the usher β-barrel domain. **b** A sketch of pilus biogenesis by the CU pathway, based on prior information and shown here with P pilus components. OM outer membrane, IM inner membrane. **c** Cryo-EM of the ΔE and ΔEF P pilus assembly intermediates revealed three conformations. The 3D map of conformer I is at a resolution of 7.6 Å, conformer II at 3.8 Å and conformer III at 7.2 Å. Subunits are individually colored in the same scheme as in (a).

PapF (PapK_{F-NTE}) and deleted *papE* and *papF*, resulting in a ΔEF P pilus tip fiber composed only of PapK bound to PapG (Supplementary Fig. 1b). Bacteria expressing P pili with the modified tip fibers agglutinated red blood cells similar to bacteria expressing wild-type P pili (Supplementary Fig. 1c), demonstrating that the ΔE and ΔEF tips are biologically active and functional.

**Cryo-EM reveals three conformations.** The in vivo-produced PapCDK_{E-NTE}FG (ΔE) and PapCDK_{F-NTE}G (ΔEF) P pilus assembly intermediates were purified from the OM in the detergent n-dodecyl β-D-maltoside (DDM) using a His-tag appended to the PapD chaperone (Supplementary Fig. 2a, b).

Twenty-five thousand one hundred and eighty-three micrographs of ΔE and 14,722 micrographs of ΔEF were recorded on a K2 detector in a 300 kV Titan Krios with an under-focus range of 1.5–2.5 μm. The raw particles of ΔE and ΔEF were homogeneous and the particle orientations were well-sampled (Supplementary Fig. 2c–f). For the three-subunit ΔE dataset, starting with 537,038 particles, after 2D and 3D classifications, we derived one 3D map, named Conformer I at 7.6 Å. For the two-subunit ΔEF dataset, starting with over one million particles, after 2D and 3D classifications, we derived two 3D maps, named Conformers II and III, at 3.8 Å and 7.2 Å, respectively. (Fig. 1c, Supplementary Figs. 3, 4, Supplementary Table 1). Crystal structures are available for

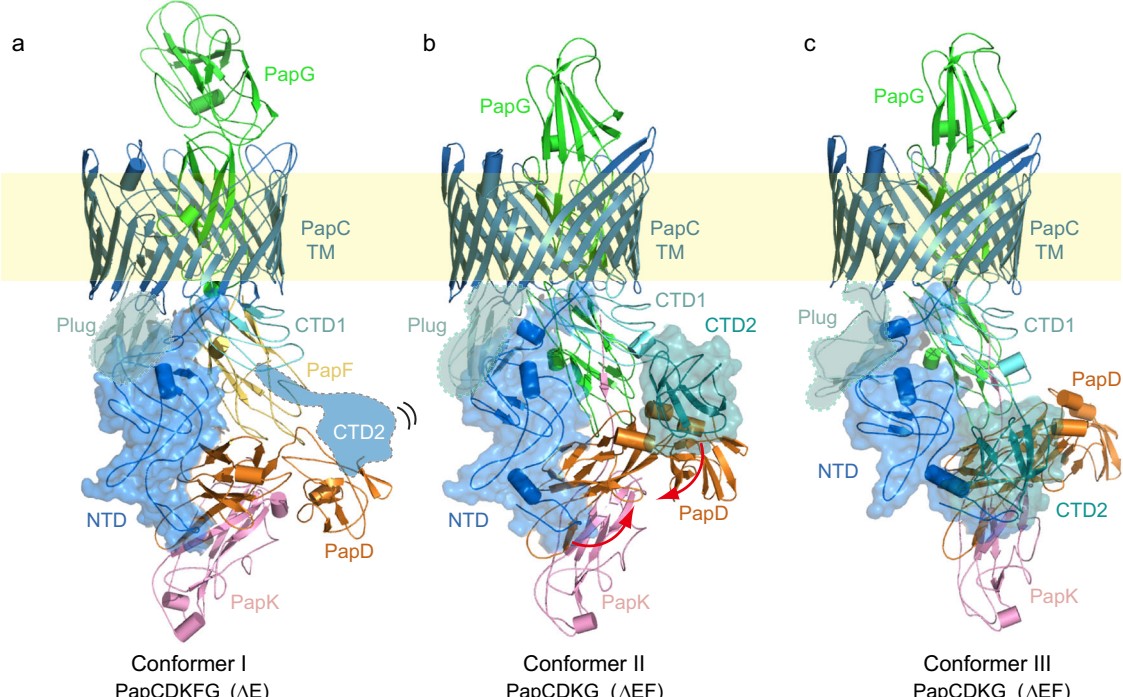

**Fig. 2 Atomic models of the ΔE and ΔEF complexes in three distinct configurations. a** Atomic model of ΔE Conformer I. **b, c** Atomic models of ΔEF Conformer II and Conformer III. The NTD and β-barrel of PapC are colored in blue; PapC plug domain in aquamarine; PapC CTD1 and CTD2 in light-teal and teal, respectively; PapD in orange; PapK in pink; PapF in yellow; and PapG in green. The flexible and invisible PapC CTD2 is highlighted in cartoon view in (**a**). The plug domain is highlighted in teal and surface views of the NTD and CTD2 are shown (**a–c**). TM, transmembrane.

each of the proteins in the Pap complexes[9,11,18,28], which allowed for atomic models to be built with confidence at these resolutions for most parts of the cryo-EM maps, with remaining parts built de novo with the help of large aromatic residues (Trp, Tyr, Phe, and His) (Fig. 2a–c, Supplementary Fig. 4). Model building for the lower resolution Conformers I and III was assisted by the rigidity of the Ig-like protein domains in the CU system and the near-atomic resolution structure obtained for Conformer II.

The ΔE cryo-EM structure (Conformer I) represents a P pilus assembly intermediate consisting of the PapC usher, PapD chaperone, and three polymerized pilus subunits: the PapG adhesin, with its pilin domain translocating through the PapC β-barrel channel and its lectin domain exposed on the extracellular side; and PapF and PapK, which are located on the periplasmic side of the usher (Figs. 1c, 2a). PapK is bound to the PapD chaperone via DSC, and PapK is also linked (by its engineered PapE NTE) to PapF via DSE. PapF in turn is engaged in DSE with the pilin domain of PapG. In the ΔEF cryo-EM structures (Conformers II and III), the PapF subunit is not present and instead PapK is directly engaged in DSE (by its engineered PapF NTE) with the pilin domain of PapG (Figs. 1c, 2b, c). The lectin domain of PapG in Conformers II and III has not yet fully transited through the usher channel and the PapG pilin domain is still located on the periplasmic side of the usher. The ΔEF conformers are the first structures obtained for a two-subunit CU pilus assembly intermediate. In Conformers II and III, the PapC plug domain, NTD, and CTDs are resolved in the 3D maps and reside on the periplasmic side of the β-barrel channel. In contrast, CTD1 is not visible at the normal display threshold in Conformer I, becoming visible only at a lower threshold, and CTD2 is missing entirely. This observation indicates that the CTDs have significant flexibility in Conformer I. In Conformer III, the arrangement of the usher NTD and CTDs is similar to their arrangement in the previously determined PapCDG crystal

structure[18]. Conformer III is also similar to a previously determined conformer of the FimDCFGH type 1 pilus tip complex[24]. In contrast, Conformers I and II capture new states of the usher during pilus assembly.

**Three-step capture of chaperone–subunit complexes by CTD2.** In the Conformer I ΔE structure, the PapD chaperone and associated PapK subunit are held on the periplasmic side of the PapC usher by the N-terminal tail region of the usher NTD (Fig. 2a). This interface between the NTD and chaperone–subunit complex was previously observed in the PapCDG crystal structure and type 1 pilus tip complex[18,24]. Notably, PapC CTD2 is not resolved in the ΔE structure, indicating that this usher domain is released from the bound chaperone-tip complex and flexible. This is in contrast to the PapCDG structure and each of the previously solved type 1 pilus assembly intermediates, where CTD2 is invariably bound to the chaperone, helping to anchor the last-recruited chaperone–subunit complex (PapDK in this case) at the usher[16,18,23,24]. Therefore, Conformer I likely represents the stage just after a newly recruited chaperone–subunit complex (PapDK), still bound at the usher NTD, has undergone DSE with the previously recruited chaperone–subunit complex (PapDF) bound at the usher CTDs. The DSE interaction between the PapK and PapF would have caused dissociation of the chaperone bound to PapF, leading to the release (unbinding) of CTD2, as in Conformer I. At this stage, the pilus fiber is anchored at the usher by the NTD staying bound to the newly incorporated PapDK complex.

The PapD chaperone is a two-lobed, boomerang-shaped structure. In the Conformer II ΔEF structure, the usher NTD is bound to the N-terminal end of the first lobe of PapD, as in Conformer I, but here CTD2 of the PapC usher is visible (Figs. 1c, 2b). CTD2 is bound to PapD in a never-before-seen binding site at the vertex of the boomerang, where CTD2 primarily interacts

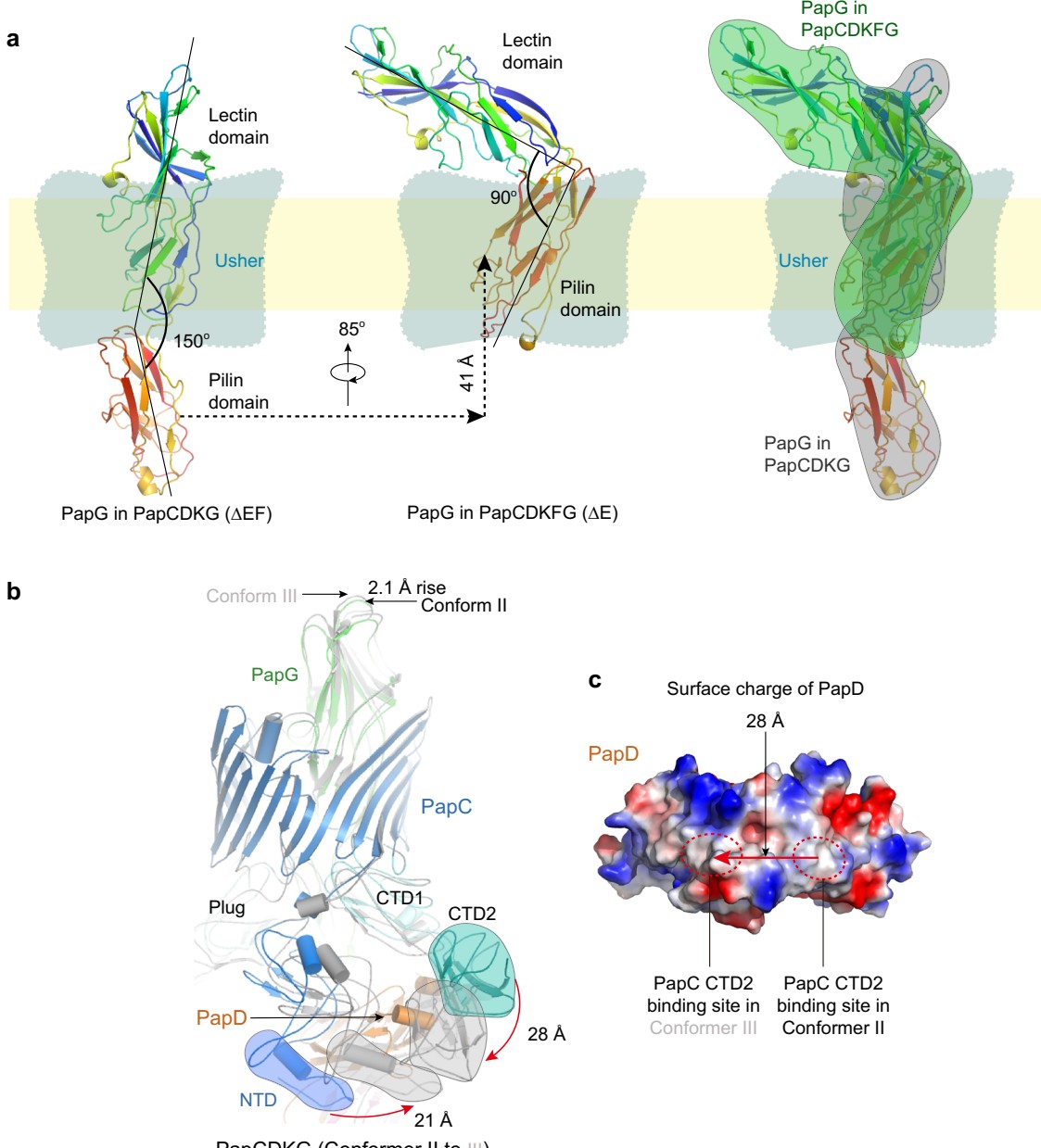

**Fig. 3 P pilus subunit rotation during secretion through the usher channel and CTD2 movement on PapD. a** Comparison of Conformers III and I shows that the PapG adhesin domain rotates as it emerges from the usher channel. **b** Comparison of Conformers II and III shows that PapC NTD and CTD2 undergo counter movements to form an NTD-CTD2 interface. **c** In the surface charged view of PapD, PapC CTD2 shifts about 28 Å from Conformer II to III, moving between hydrophobic patches on PapD. See text for details.

with the C-terminal end of the first lobe of PapD (Figs. 1c, 2b, Supplementary Fig. 5a). Due to this binding and stabilization of CTD2, we propose that Conformer II follows Conformer I in the chaperone–subunit recruitment and handover cycle that takes place on the periplasmic domains of the usher. In the Conformer III ΔEF structure, PapC CTD2 undergoes another dramatic shift, moving 28 Å toward the N-terminal end of the first lobe of PapD, next to where the usher NTD is bound, forming a PapC NTD–PapD–PapC CTD2 interface (Figs. 1–3, Supplementary Fig 5, Supplementary Movie 1). The usher's other periplasmic domains also undergo conformational changes in Conformer III compared to Conformer II (Fig. 3b). The PapC NTD moves significantly away from the plug domain, shifting laterally by ~21 Å. The movements of PapC NTD and CTD2 cause the bound PapDK to move laterally, leading to a slight upward movement of

the PapG adhesin. Importantly, CTD2 and PapDK move conversely during the shift from Conformer II to Conformer III (Fig. 3b, Supplementary Movie 1), which suggests that the usher NTD conveys the newly recruited chaperone–subunit toward the CTDs and that CTD2 assists this process through the two-step binding on PapD. The surface of the PapD chaperone is largely hydrophilic. However, the two patches where PapC CTD2 binds are hydrophobic (Fig. 3c). This structural feature may facilitate the sliding of CTD2 between the two sites on the PapD chaperone during subunit handover from NTD to CTD2. The upward movement of PapG from Conformer II to III (Fig. 3b), together with the shift of PapDK toward the CTDs side of the usher, suggests that Conformer III follows Conformer II within the subunit assembly cycle.

In terms of extension of the pilus fiber, the two-subunit ΔEF Conformers II and III are of course one assembly cycle earlier than the three-subunit ΔE Conformer I. Transitioning from the ΔEF (PapCDKG) structures to the ΔE (PapCDKFG) structure (Fig. 2) involves a rotation of the pilus tip subunits around the channel axis accompanying their outward movement through the usher channel. The N-terminal lectin domain and C-terminal pilin domain of the PapG adhesin are almost linear with an angle of 150° in the ΔEF structures. Comparison with the ΔE Conformer I shows that as the third subunit is polymerized from below, PapG moves upward (outward) by 41 Å while rotating by 85°, which is equivalent to approximately 2° rotation per 1 Å advance. Notably, as the PapG lectin domain emerges from the usher chamber, it bends back by 60° toward the pilin domain such that the two lobes of PapG are now nearly orthogonal to each other (Fig. 3a, Supplementary Movie 2). The type 1 pilus system lacks structures that are one assembly cycle apart. Nevertheless, based on molecular dynamics calculation, it was predicted that the type 1 pilus subunits rotate by 2° per 1 Å advance through the FimD usher channel[23]. Therefore, the predicted precession of a Fim subunit during secretion matches the Pap subunit precession as experimentally observed in the P pilus structures (Fig. 3a).

The N-terminal tail region of the NTD is critical for interactions of the usher with chaperone–subunit complexes and the assembly of both the P and type 1 pili[17,18,24,26,29]. As observed in our P pilus assembly intermediates, and consistent with the previous PapCDG crystal structure, PapC N-terminal tail residues F3, I16, F18, F21 interact closely with a hydrophobic pocket on the PapD chaperone including residues P30, L32, P94, and P95 (Supplementary Fig. 5a). Mutations to these N-terminal tail residues ablate or impair pilus biogenesis[18,26,29]. Our structures reveal additional hydrophobic contacts between the usher NTD and chaperone: L28 and Y32 of PapC interact with P54, P55, and V56 of PapD (Supplementary Fig. 5a). PapC L28A and Y32A point mutants partly or totally disrupt P pilus biogenesis, as measured by hemagglutination assay (Supplementary Fig. 5c). The interactions between PapC CTD2 and PapD are more complicated: in Conformer II, F732 and W767 of CTD2 interact hydrophobically with A119 and I120 of PapD, respectively; in Conformer III, F732 and W767 of CTD2 form hydrophobic interactions with I46, I51, and P73 of PapD (Supplementary Fig. 5a, b). Consistent with an important functional role for these CTD2-chaperone interactions, PapC F732A, W767A, and L765A/W767A point mutants disrupt P pilus biogenesis (Supplementary Fig. 5c). In Conformer III, CTD2 also makes direct contact with the N-terminal tail region of the NTD: CTD2 residues P744, F745, and W781 interact with NTD residues L8 and I16 in a hydrophobic network (Supplementary Fig. 5b). As shown in the previous PapCDG study, mutations to this NTD-CTD2 interface disrupt pilus biogenesis[18]. The residues newly revealed by our structures to be involved in contacts between the PapC periplasmic domains and PapD chaperone are located in regions that are highly conserved among members of the usher superfamily (Supplementary Fig. 6).

The NTD–chaperone–CTD2 interface observed in Conformer III was previously visualized for the type 1 pilus tip complex and the PapCDG crystal structure[18,24] (Supplementary Fig. 7). However, in the PapCDG structure, the usher is in the inactive, closed state and the plug domain is still inside the ß-barrel channel (Supplementary Figs. 8d, 9). An important difference between the previously resolved type 1 pilus assembly intermediates and the P pilus structures is the distance between subunits docked at the usher CTDs and the periplasmic face of the usher ß-barrel channel. In the type 1 pilus complexes, a single subunit domain is accommodated below the periplasmic face of the usher, whereas two domains are accommodated below the entrance to the usher channel in each of the P pilus complexes (Fig. 2, Supplementary Fig. 8). This difference provides an explanation for why recruitment of the two-domain PapG adhesin alone does not activate the PapC usher. As shown by our ΔEF structures, recruitment of an additional subunit (PapK in this case) is then sufficient to gate open and activate the usher. Comparison of ΔEF Conformer III with the PapCDG crystal structure reveals significant additional changes that likely reflect requirements for the expulsion of the plug domain and activation of the usher. Movement of the plug from the usher channel to the periplasm results in a shift of the NTD away from the pilus fiber and out from under the channel entrance, with the plug taking a position near the site formerly occupied by the NTD (Supplementary Fig. 9). A plug–NTD interface is formed by interactions involving hydrophobic and charged residues (plug residues P254, R256, D297; NTD residues Y128, D130, W133) (Supplementary Fig. 9). Consistent with an important functional role for this interface, residue R256 forms part of a charge-pair network previously shown to contribute to channel gating and P pilus biogenesis[30]. These changes in the usher domains allow a straightening and realignment of the pilus fiber under the usher channel, with the channel itself changing from reniform to round (Supplementary Figs. 8b, d, 9c). However, a direct pathway for the expulsion of the plug from the closed usher as shown in the PapCDG crystal structure[18] to reach the active and open state as shown in our in vivo assembly intermediates is not apparent, as the plug and lectin domain of PapG would clash. Additional studies will be needed to determine how this transition occurs.

## Discussion

We present here the first structures of the activated PapC usher during P pilus biogenesis. The P pilus structures in this study show that the usher CTD2 adopts three distinct states during pilus assembly: released and flexible in Conformer I; bound at the vertex of the PapD chaperone in Conformer II; and shifted to the N-terminal arm of PapD in Conformer III. These CTD2 changes are countered by shifts in the usher NTD, conveying the bound chaperone–subunit complex toward the CTDs side of the usher in Conformer III. Our previous analysis of the type 1 pilus tip complex revealed a folding-unfolding cycle of the N-tail region of the usher NTD[24]. Thus, coordinated conformational cycles of the usher N- and C-terminal periplasmic domains function to drive successive rounds of chaperone–subunit recruitment, incorporation into the pilus fiber, and secretion through the usher channel to the cell surface.

Our data, together with prior findings, suggest the following sequential model for P pilus biogenesis at the PapC usher, starting with the PapCDG crystal structure[18] (Fig. 4). (1) Recruitment of the initiating PapDG chaperone–adhesin complex to the usher is insufficient to gate open the usher channel, as PapC accommodates two-subunit domains below its channel entrance. Based on studies of FimD[24], the usher NTD exists in an equilibrium between a bound state, as observed in the PapCDG crystal structure and Conformer III, and a "recruitment" state where the NTD is released and the N-terminal tail is disordered. (2) Upon recruitment of the next chaperone–subunit complex (PapDF) by the PapC NTD, the N-terminal tail folds into a helical motif[17] and PapF is positioned on the usher NTD to undergo DSE with the previously recruited PapDG complex bound at the CTDs. In a process not yet understood, the usher channel is activated, leading to the release of the plug domain into the periplasm and insertion of the PapG lectin domain into the channel. (3) DSE between the PapF and PapG forms the first link in the pilus fiber and also drives the release of the PapD chaperone from PapG, coupled

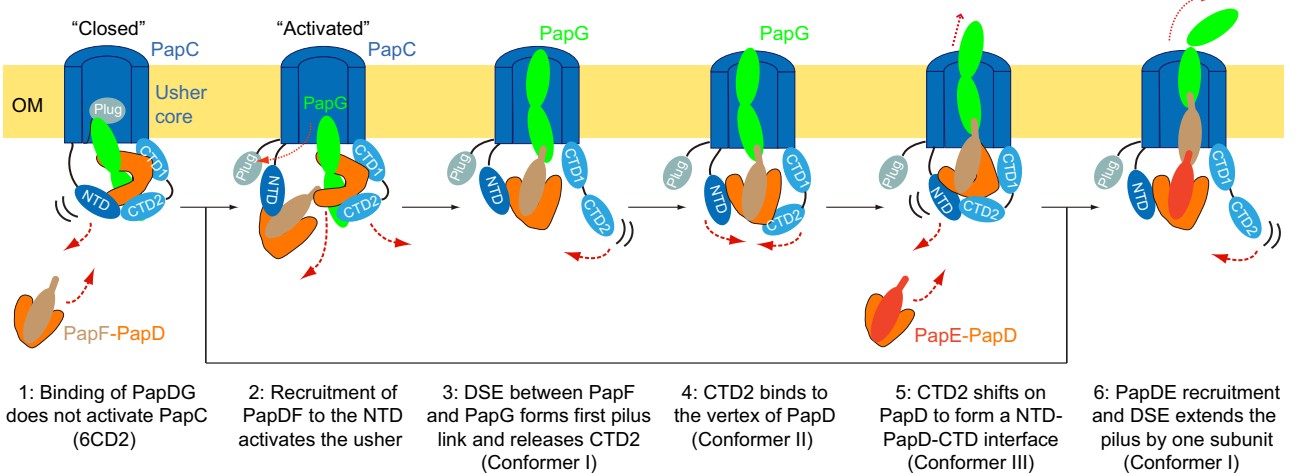

**1: Binding of PapDG does not activate PapC (6CD2)**

**2: Recruitment of PapDF to the NTD activates the usher**

**3: DSE between PapF and PapG forms first pilus link and releases CTD2 (Conformer I)**

**4: CTD2 binds to the vertex of PapD (Conformer II)**

**5: CTD2 shifts on PapD to form a NTD-PapD-CTD interface (Conformer III)**

**6: PapDE recruitment and DSE extends the pilus by one subunit (Conformer I)**

**Fig. 4 Sequential model for P pilus biogenesis at the PapC usher.** The illustration highlights the proposed assembly pathway, starting with the PapCDG crystal structure (6CD2) and moving through the three distinct ΔE and ΔEF conformations. See text for details. OM, outer membrane.

with the release of CTD2 (Conformer I). The PapC NTD maintains its hold on the subunit, with the N-terminal tail rearranging into a loop motif[24]. (4) Next, the handover of PapDF from the NTD to the CTDs is initiated by binding of CTD2 to the midpoint of the PapD chaperone (Conformer II). (5) CTD2 then shifts toward the N-terminal end of the first lobe of PapD, forming an NTD–chaperone-CTD2 interface, which is accompanied by a counter movement of the NTD and bound chaperone–subunit complex toward the CTDs (Conformer III). Handover of the newly incorporated PapDF complex to the CTDs is driven by the higher affinity of the CTDs compared to the NTD[20], as well as the formation of the NTD-CTD2 interface, which destabilizes NTD binding to the chaperone[19]. Completion of the handover process releases the NTD, thereby resetting the usher for a new cycle of chaperone–subunit (PapDE) recruitment and incorporation (6), leading to the outward translation and rotation of the pilus fiber through the usher channel[23]. Additional cycles then allow for polymerization and secretion of the pilus tip and rod.

In conclusion, the structures presented in this study reveal new conformational dynamics of the pilus assembly process, highlight differences between the P and type 1 pilus biogenesis, and show how the usher periplasmic domains cooperate to drive extension and outward secretion of the pilus fiber through the usher channel to the cell surface. These structures will pave the way for novel therapeutics designed against P pili, as well as other virulence factors assembled by the CU pathway, which may provide alternatives to the use of antibiotics in the treatment of urinary tract infections and other infectious diseases.

## Methods

**Strains and plasmids.** The bacterial strains and plasmids used in this study are listed in Supplementary Table 2. Bacteria were grown at 37 °C with aeration in LB medium with appropriate antibiotics. *E. coli* DH5α was used as the host strain for plasmid manipulations and all constructs were sequenced to verify the intended mutations. Plasmids pGW990 and pGW992, encoding PapD$_{His}$K$_{E-NTE}$FG (ΔE) and PapD$_{His}$K$_{F-NTE}$G (ΔEF), respectively, were constructed by site-directed, ligase-independent mutagenesis (SLIM)[31,32] using the primers listed in Supplementary Table 3. Starting with plasmid pJL01, encoding PapD$_{His}$JKEFG, SLIM was used to delete *papE* or *papEF*. Then, SLIM was used again to replace the PapK NTE with the NTE of PapE (ΔE) or PapF (ΔEF). For construction of the PapC point mutants, plasmid pDG2 (PapC$_{His}$) was mutated using the QuikChange Site-Directed Mutagenesis Kit (Stratagene) and the primers listed in Supplementary Table 3, generating plasmids pNH577 (PapC L28A), pNH580 (Y32A), pNH583 (F732A), pNH584 (W767A) and pNH585 (L765A/W767A). Proper expression and folding of the PapC mutants in the bacterial OM were determined in strain SF100 by a heat-modifiable mobility assay[33]. OM fractions were isolated by Sarkosyl extraction, and the samples were incubated in SDS sample buffer for 10 min at 25 or

95 °C prior to separation by SDS-PAGE and immunoblotting with anti-His-tag antibody (BioLegend) at 1:1000 dilution.

**PapC-tip complex expression and purification.** Plasmids pGW990, encoding PapD$_{His}$K$_{E-NTE}$FG, and pKD101, encoding PapC, were transformed into *E. coli* Tuner competent cells to overexpress the ΔE complex PapCD$_{His}$K$_{E-NTE}$FG. Similarly, plasmids pGW992, encoding PapD$_{His}$K$_{F-NTE}$G, and pKD101 were transformed into *E. coli* Tuner competent cells to overexpress the ΔEF complex PapCD$_{His}$K$_{F-NTE}$G (ΔEF). 18 L cultures of bacteria expressing the ΔE or ΔEF complexes were grown at 37 °C with aeration in LB supplemented with 50 μg/mL kanamycin and 100 μg/mL ampicillin. At OD$_{600}$ = 1.0, the cultures were induced with 50 μM IPTG for 3 h at 37 °C. The bacteria were harvested and disrupted with a microfluidizer. The disrupted cells were spun (10,000 × g, 20 min, 4 °C) to separate cellular debris and unbroken cells. 0.5% (w/v) Sarkosyl was added to the supernatant to solubilize the inner membrane (stirring, 20 min, room temperature). The OM was isolated by ultracentrifugation (100,000 × g, 50 min, 4 °C). The OM was resuspended using 25 mM Tris-HCl (pH 8.0), 300 mM NaCl, 15% glycerol, 10 mM MgCl$_2$, and protease inhibitors. The ΔE and ΔEF protein complexes were solubilized (stirring, overnight, 4 °C) using 25 mM Tris-HCl (pH 8.0), 300 mM NaCl, and 1% (w/v) n-Dodecyl β-D-maltoside (DDM; Anatrace). The insoluble material was spun out by ultracentrifugation (100,000 × g, 1 h, 4 °C). The solubilized ΔE and ΔEF complexes were applied to a Ni-NTA column and eluted using buffer 25 mM Tris-HCl (pH 8.0), 300 mM NaCl, 250 mM imidazole, and 0.1% DDM. The purified ΔE and ΔEF complexes were concentrated using a Centricon 100 concentrator. The concentrated samples were applied to a Superdex 200 size-exclusion chromatography column (GE Healthcare) equilibrated with 25 mM Tris-HCl (pH 8.0), 120 mM NaCl, and 0.1% DDM. The fractions off the size-exclusion chromatography were analyzed by SDS-PAGE (Supplementary Fig. 2a, b). The fractions containing the ΔE and ΔEF complexes were concentrated to a final concentration of 2.4 mg/mL.

**Cryo-EM.** To prepare cryo-EM grids, we applied 2 μL of the purified samples to glow-discharged C-flat 1.2/1.3 holey carbon grids, incubated for 10 s at 6 °C and 95% humidity, blotted for 3 s then plunged into liquid ethane using an FEI Vitrobot IV. In C-flat R1.2/1.3 holey carbon film grids, the PapCD$_{His}$K$_{E-NTE}$FG (ΔE) and PapCD$_{His}$K$_{F-NTE}$G (ΔEF) particles distributed well with no aggregation problem. We loaded the grids into an FEI Titan Krios electron microscope operated at a high tension of 300 kV and collected images semi-automatically with Serial-EM under low-dose mode at a nominal magnification of ×130,000 and a pixel size of 1.029 Å per pixel. A Gatan K2 Summit direct electron detector was used under super-resolution mode for image recording with an under-focus range from 1.5 to 2.5 μm. A Bioquantum energy filter installed in front of the K2 detector was operated in a zero-energy-loss mode with an energy slit width of 20 eV. The dose rate was 10 electrons per Å$^2$ per second and the total exposure time was 6 s. The total dose was divided into a 30-frame movie with a dose per frame of 2 e$^-$/A$^2$.

**Image processing and 3D reconstruction.** We collected 14,722 raw movie micrographs of the ΔEF complexes and 25,183 raw movie micrographs of the ΔE complex. The movie frames were first aligned and superimposed by the program Motioncorr 2.0[34]. Contrast transfer function parameters of each aligned micrograph were calculated using the program CTFFIND4[35]. All the remaining steps, including particle auto selection, 2D classification, 3D classification, 3D refinement, and density map postprocessing were performed using Relion-3.0[36]. For the ΔEF

complex, the template for automatic picking was generated from a 2D average of about ~10,000 manually picked particles in different views. Automatic particle selection was performed for the entire dataset, and 1,054,098 particles were initially picked. We then carefully inspected the selected particles, removed "bad" ones and re-picked some initially missed "good" ones, and sorted the remaining good particles by similarity to the 2D references, in which the bottom 10% of particles with the lowest z-scores were removed from the particle pool. 2D classification of all good particles was performed and particles in the classes with unrecognizable features by visual inspection were removed. A total of 598,413 particles were used for further 3D classification. We derived five 3D models from the dataset and chose the two best models for the final refinement (Supplementary Fig. 3). The other three models were distorted and those particles were discarded. The final two datasets have 227,396 and 131,650 particles respectively. They were used for further 3D refinement, resulting in 3.8 Å and 7.2 Å 3D density maps. For the ΔE complex, we used a similar process as for the ΔEF complex. After 2D classification, 239,364 particles were used for further 3D classification. We derived five 3D models from the dataset and combined three similar models for the final refinement. The other two models were distorted and those particles were discarded. The final dataset comprised 119,682 particles and was used for further 3D refinement, resulting in a 7.6 Å 3D density map. The resolution of each map was estimated by the gold-standard Fourier shell correlation, at the correlation cutoff value of 0.143. The 3.8 Å, 7.2 Å, and 7.6 Å density maps were sharpened by applying a negative B-factor of −115, −276, and −229 $Å^2$, respectively (Supplementary Fig. 4).

**Atomic modeling, refinement, and validation**. The X-ray crystal structures of the PapCDG complex (PDB ID 6CD2), PapDK (1PDK) and PapF (2W07) were used as the initial model. In Conformer II, the model of the PapC core was initially built using the Swiss model online server (https://swissmodel.expasy.org/), and the generated model was subsequently adjusted manually in COOT guided by residues with bulky side chains like Arg, Phe, Tyr, and Trp. The NTD, CTD, and plug domains of PapC, and PapG of PapCDG (6CD2) were separately fitted as rigid bodies into the EM density of Conformer II and further manually refined in COOT. The PapDK crystal structure (1PDK) was docked as a rigid body into the EM map in Chimera and COOT. For Conformer III modeling, the previous Conformer II model was initially docked into the EM map as a rigid body. Then, the CTD2 domain was shifted to the NTD side of PapD and docked into the corresponding density. For Conformer I, PapC, PapG, and PapDK of Conformer II were docked separately as rigid bodies into the Conformer I EM map. PapF (PDB ID 2W07) was further fitted into the density between PapG and PapK. These models were refined by rigid body refinement of individual chains in the PHENIX program[37], and subsequently adjusted manually in COOT. Finally, the atomic models were validated using MolProbity[38]. Structural figures were prepared in Chimera and Pymol (https://www.pymol.org).

**Hemagglutination (HA) assay**. The assembly of functional P pili on the bacterial surface was determined using a HA assay[33]. For comparison of wild-type P pili with bacteria expressing the ΔE and ΔEF tip constructs, strain AAEC185/pKD101 (PapC) + pTN46 (PapA) was transformed with plasmid pJL01 (PapD-HisJKEFG), pGW990 (ΔE, PapDHisKE-NTEFG) or pGW992 (ΔEF, PapDHisKF-NTEG). For comparison of wild-type PapC with the PapC point mutants, strain AAEC184/pMJ2 (PapAHDJKEFG) was transformed with pDG2 (WT PapC), pNH577 (PapC L28A), pNH580 (PapC Y32A), pNH583 (PapC F732A), pNH584 (PapC W767A) or pNH585 (PapC L765A/W767A). HA titers were performed by serial dilution of the bacteria in microtiter plates and recorded visually as the greatest fold dilution of bacteria able to agglutinate human red blood cells. Red blood cells were obtained with informed consent according to a protocol approved by the Institutional Review Board of Stony Brook University. HA titers were calculated from at least three independent experiments, with three replicates per experiment.

**Reporting Summary**. Further information on research design is available in the Nature Research Reporting Summary linked to this article.

## Data availability
The cryo-EM 3D maps of the PapC-tip complex generated in this study have been deposited in the EMDB database with accession codes EMD-23341 (Conformer I), EMD-23339 (Conformer II) and EMD-23340 (Conformer III). The corresponding atomic models were deposited at the RCSB PDB with accession codes 7LHI (Conformer I), 7LHG (Conformer II), and 7LHH (Conformer III).

RCSB PDB accession codes for previously reported structures cited in this study are as follows: PapCDG (6CD2), PapDK (1PDK), PapF (2W07), FimDCFGH (6E14), FimDCH (3RFZ).

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

## Acknowledgements

Cryo-EM data was collected at the David Van Andel Advanced Cryo-Electron Microscopy Suite at the Van Andel Institute. We thank Dr. Xing Meng for help with data collection. This study was supported by US National Institutes of Health grants R01 GM062987 (to D.G.T.) and R35 GM131754 (to H.L.), and by the Van Andel Institute (to H.L.). G.T.W. was supported by US National Institutes of Health awards T32 GM008444 and F30 AI112252.

## Author contributions

H.L., D.G.T., and G.T.W. conceived and designed experiments. M.D., G.T.W., N.H.S., H.C., A.K., and J.J. performed biochemical and molecular biology experiments. M.D., Z.Y., and G.Z. performed cryo-EM experiments. M.D. and Z.Y. performed image processing and atomic modeling. M.D., Z.Y., G.T.W., H.L., and D.G.T. analyzed the data. M.D., Z.Y., G.T.W., H.L., and D.G.T. wrote the manuscript.

## Competing interests

The authors declare no competing interests.
