## [Peer Review File · Nature Communications]

Processive Dynamics of the Usher Assembly Platform During Uropathogenic Escherichia coli P Pilus BiogenesisREVIEWER COMMENTS

Reviewer #1 (Remarks to the Author):

Review: Processive Dynamics of the Usher Assembly Platform During Uropathogenic Escherichia coli P Pilus Biogenesis

This study uses cryo-EM coupled with a clever in vivo expression system to capture three intermediates states during P pilus assembly. The method devised eliminated many issues that previously plagued cryoEM sample preparation, including heterogeneity (due to variable number of PapE subunits) and pilus flexibility. Using the PapCDK(NTE of F)G and PapCDK(NTE of E)FG constructs, the authors were able to provide insight into the molecular architecture of the PapC usher secreting two and three subunits.

The manuscript is well presented, and the authors are able to make important contributions to our understanding P pilus assembly at the PapC usher.

The major concerns are as follows: (1) Two of the three conformations (Conformers I and III, which represent PapCDKFG and PapCDKG, respectively) were solved at relatively low resolution (7.6 and 7.2 angstroms, respectively) which introduces ambiguity in the placement of individual domains. The authors relied on previously-solved structures to fit the density maps, which may be problematic, especially when those structures were from different steps in the assembly process (2) While the constructs expressed in E. coli were able to produce functional pili as shown by hemagglutination assay, it remains unclear if the re-arrangement of subunits resulted in different conformations than what would be observed in wildtype pili and, in particular, if inferences into the PapC periplasmic domain movements may still be drawn. For example, the PapCDK(NTE of F)G may not be an appropriate comparison to PapCDFG; Otherwise, one is lead to believe PapF and PapE serve no functional role, in which case the question of why the Pap operon has evolved to have both PapF and PapE arises.

The minor concerns are as follows: (1) Figure 1b claims to show a sketch of P pilus biogenesis, yet it may actually be a more accurate depiction for type 1 pili, since many of these early steps in usher activation and pilus biogenesis remain debated for P pili. For example, it is not known if movement of PapDF from the NTD to the CTD requires recruitment of the next subunit (PapE). Later in the text, the authors make an unsubstantiated claim that the ΔEF structure show that recruitment of PapF would be sufficient to remove the plug domain and activate the usher, but this has not been empirically shown. (2) PapC CTD2 is not resolved in the ΔE structure, and the authors interpret this as meaning the usher domain is released from the bound chaperone-tip complex. Alternative explanations should be explored, as this is just one interpretation that is inferred simply from lack of structure (3) The authors present an interesting finding that the PapC CTD2 is bound to PapD at the vertex of the boomerang; However, this novel finding should be supported by figures displaying specific interactions between key amino acids. Mutational analysis may aid in elucidating how this interaction occurs. (4) The authors state that the NTD "pushes" the newly recruited chaperone-subunit towards the CTDs, while it is not shown if this is simply synchronized movement driven by some other factor, or if the NTD indeed exerts a "pushing" force. (5) The authors claim there is "upward movement" when comparing the location of PapG in Conformers II and III (Fig 3b), but the structures presented seem to show no difference in the location of PapG relative to the PapC usher.

Reviewer #2 (Remarks to the Author):

Summary

This article by Du et al., describes three new cryo-EM reconstructions of the activated PapC usher in process of secreting two (PapCDKG; ΔEF – in two conformations) and three (PapCDKG; ΔE) subunits. The single ΔE reconstruction is denoted as conformer I, while the two ΔEF reconstructions are referred to as conformers II and III. These three new structures of assembly

intermediates, together with previous structural data, allow the authors to propose a model of P pilus assembly via the PapC usher. In terms of what is happening with the periplasmic usher domains and incoming chaperone-subunit complexes, the author propose an assembly order progressing from conformer I → II → III. Interestingly, this reveals a novel interface between CTD2 and the middle (vertex) of PapD (conformer II) which then rearranges to rebind further towards the N-terminal end of the first PapD lobe (conformer III), forming an interface between the NTD, PapD and the CTD2. Conversely the authors propose a progression from conformers II/III towards conformer I in terms of pilus fibre extension, given that the latter is one assembly cycle ahead in terms of the number of subunits that have been incorporated. I think this work represents an important addition of structural information to the field. The manuscript is well and clearly written, with beautifully presented figures. I have a few minor suggestions/questions, but support publication.

Minor points

Lines 32 and 88: I would suggest that molecular mechanisms of pilus assembly and function could rather facilitate development of "pilus assembly-targeted" rather than "pilus-targeted" therapeutics. Structures of the pilus fibres may lead to "pilus-targeted" therapeutics.

Lines 36/37: Perhaps you could insert the following reference: M. D. Crespo et al., "Quality control of disulfide bond formation in pilus subunits by the chaperone FimC," *Nat. Chem. Biol.*, vol. 8, no. 8, pp. 707–713, Aug. 2012.

Lines 70-71: For completion, could include references to pilus fibre structures?

Line 145: Consider rephrasing "dynamics".

Line 160: Perhaps consider removing "observed" and simply say "as in Conformer I", to avoid any confusion about the fact that no CTD2 density is visible in this state.

Line 173: Are all figure panels referred to here relevant for this statement?

Line 223: This statement could "suggest" that all of these residues are conserved, which is not entirely true. Slight rephrasing? What is "O" in the alignment shown in Extended Data Fig. 6?

Line 244: By listing all interacting residues in the bracket, it is unclear which belong to the plug and which to the NTD.

Lines 252-253: The observation that there is no direct path for the plug from the "closed" to the "open" state, does not really suggest that the "closed" state structure alone requires further validation.

Line 353/Extended Data Table 1: The total electron dose in the table is 50 e-/Å², the text suggests 60 e-/Å².

Line 354: Rather than stating how long each frame was exposed for, perhaps state the dose/frame.

Line 361/Extended Data Table 1: Typo? Relion-3.0?

Lines 374-376: Question. Did you try to individually refine these three 3D classes? Or class 1 and 3 combined? Are these three combined classes really similar enough and could this explain the modest resolution of Conformer I?

Figures

Figure 1c: Could you please label CTD1 in conformer III?

Extended Data Fig. 5: I46 mentioned in line 216 is not indicated on the figure. What do the red

asterisks indicate? Residues chosen for mutation? Mention in legend.

Extended Data Fig. 6: What is "O"

Reviewer #3 (Remarks to the Author):

Minge Du, Zuanning Yuan, Glenn T. Werneburg et al. present new structures of the P pilus assembly intermediates of the Chaperone-Usher pathway, ingeniously obtained after chimeric construction of subunit PapK with substituted N-terminal extension of PapE or PapF. The outer-membrane transporter (PapC), the chaperone (PapD) and the subunits (PapK, PapG and PapF) are co-expressed and co-purified from engineered *E. coli*. As a result, they managed to obtain cryo-EM structures of complex PapCDKFG (named ΔE or conformation I) and two different conformations (II and III) of the complex PapCDKG (named ΔEF).

Despite the low resolution of the cryo-EM structures, the authors managed to describe, for the first time, the conformation of the activated PapC usher and to highlight details of the handover mechanism of chaperone-subunit complex from NTD to CTD2. Besides, the authors described the differences between type 1 pilus and P pilus activation mechanism of the usher, i.e. activation of the channel gate and initial translocation of the subunit. Altogether, the authors proposed a new model for P pilus formation at the initial stage of the biogenesis. It is a significant result in the field of structural biology of pilus biogenesis of the chaperone-usher pathway.

Overall the manuscript is well written. I would suggest additional modifications for validity and significance of the data:

1. Extended data figure 4d: It seems that there is no helical fold, but only sheets.
2. Extended data figure 5b: because of the low resolution of conformer III, it would be more convincing to show the density map of the residues involved in the interface between PapC-NTD and -CTD2.
3. Extended data figure 5c: regarding the point mutations, a lower HA titer could result from less stable, rather than less functional usher. For instance, western-blot of PapC mutant would help to show normal expression and addressing to the outer-membrane.
4. Manuscript line 222-224: except Y32, most of the residues of the interacting interface do not seem to be conserved according to extended data figure 6.
5. Extended data figure 6: it would be interesting to add bacterial species of the different usher proteins.

Nature Communications manuscript NCOMMS-21-02282-T

Processive Dynamics of the Usher Assembly Platform During Uropathogenic *Escherichia coli* P Pilus Biogenesis

REVIEWER COMMENTS

Reviewer #1

Major Concerns:

Point 1: Two of the three conformations (Conformers I and III, which represent PapCDKFG and PapCDKG, respectively) were solved at relatively low resolution (7.6 and 7.2 angstroms, respectively) which introduces ambiguity in the placement of individual domains. The authors relied on previously-solved structures to fit the density maps, which may be problematic, especially when those structures were from different steps in the assembly process

Response: This is a fair criticism, as interpretation of a map at 7–8 Å resolution often introduces ambiguity in domain assignment. However, this is less of a concern in our particular system, for two reasons. (1) Most domains of the chaperone-usher system are Ig-like folds with β -sheet cores; they are rock solid and the domains themselves do not undergo significant conformational changes in the different pilus assembly states. This observation is supported by numerous structural studies of the type 1 pilus, P pilus and other chaperone-usher systems. (2) We do have a near atomic resolution structure of Conformer II, which reveals the individual domain structures. Our domain placement took advantage of the rigidity of these domains and the higher resolution of Conformer II. We have added this information to the revised manuscript (lines 134-137).

Point 2: While the constructs expressed in *E. coli* were able to produce functional pili as shown by hemagglutination assay, it remains unclear if the re-arrangement of subunits resulted in different conformations than what would be observed in wildtype pili and, in particular, if inferences into the PapC periplasmic domain movements may still be drawn. For example, the PapCDK(NTE of F)G may not be an appropriate comparison to PapCDFG; Otherwise, one is lead to believe PapF and PapE serve no functional role, in which case the question of why the Pap operon has evolved to have both PapF and PapE arises.

Response: Pilus biogenesis at the usher is a modular and iterative process. From the perspective of assembly of the pilus fiber, the pilin domains of each subunit form equivalent structural units, with each subunit's N-terminal extension (NTE) and acceptor groove determining the assembly order of the pilus fiber. We have modified the text on lines 105-108 of the revised manuscript to emphasize these points. It is this modular nature of pilus assembly that allowed us to generate the P pilus assembly intermediates reported in our study. The Δ EF and Δ E fibers we generated are directly comparable to each other and are valid representations of two- and three-subunit P pilus assembly intermediates. However, in no way do our structures comment on the functional roles of the PapF and PapE subunits in the native P pilus fibers. The different chaperone-usher pili are highly evolved to their specific functions. For example, type 1 pili are adapted to bind mannosylated protein receptors under shear stress in the bladder, whereas P pili are adapted to bind glycolipids in the lower shear environment of the kidney; these pili have very different tip structures and binding modes. In fact, our new pilus constructs now provide a means to explore the physiological role of pilus architecture during infection of the host.

Minor Concerns:

Point 1-1: Figure 1b claims to show a sketch of P pilus biogenesis, yet it may actually be a more accurate depiction for type 1 pili, since many of these early steps in usher activation and pilus biogenesis remain debated for P pili. For example, it is not known if movement of PapDF from the NTD to the CTD requires recruitment of the next subunit (PapE).

Response: The legend for **Fig. 1b** has been changed as follows: “**b**, A sketch of pilus biogenesis by the CU pathway, based on prior information and shown here with P pilus components.”

Point 1-2: Later in the text, the authors make an unsubstantiated claim that the ΔEF structure show that recruitment of PapF would be sufficient to remove the plug domain and activate the usher, but this has not been empirically shown.

Response: The previous one-subunit PapCDG crystal structure showed that the usher is in a closed state. Our two-subunit ΔEF structure shows that the usher is now activated with the plug expelled and the PapG adhesin inserted into the channel. Therefore, we can conclude that recruitment of an additional, second, subunit is sufficient to gate open and activate the usher. In native pili, this second subunit would be PapF, and there is evidence from prior studies supporting a role for PapF in activating the usher (see lines 78-81 of the manuscript). However, we agree that direct support for this statement is lacking. Therefore, we have modified the manuscript (lines 251-253) to remove mention of PapF, as follows: “As shown by our ΔEF structures, recruitment of an additional subunit (PapK in this case) is then sufficient to gate open and activate the usher.”

Point 2: PapC CTD2 is not resolved in the ΔE structure, and the authors interpret this as meaning the usher domain is released from the bound chaperone-tip complex. Alternative explanations should be explored, as this is just one interpretation that is inferred simply from lack of structure

Response: We believe a lack of density is a reflection of the flexibility of the underlying structure, which in turn is a result of a lack of stabilizing interaction. This interpretation is consistent with previous studies demonstrating that the NTD, CTD1 and CTD2 of the usher are all flexible in the apo structure in the absence of interaction with a chaperone-subunit complex [Remaut *et al. Cell* **133**, 640-652 (2008); Phan *et al. Nature* **474**, 49-53 (2011)]. An alternative explanation would include the *in vitro* degradation of CTD2 by contaminating protease(s). However, this is highly unlikely because CTD2 is fully present in the other conformers.

Point 3: The authors present an interesting finding that the PapC CTD2 is bound to PapD at the vertex of the boomerang; However, this novel finding should be supported by figures displaying specific interactions between key amino acids. Mutational analysis may aid in elucidating how this interaction occurs.

Response: **Supplementary Fig. 5a** shows the specific interactions in Conformer II (ΔEF) between PapC CTD2 and PapD. The text has been updated to cite this figure earlier, on line 181: “(**Fig. 1c, 2b, Supplementary Fig. 5a**).” In addition, we have revised **Supplementary Fig. 5a-b** to include superimposed 3D density in the zoomed windows showing the specific

interactions. **Supplementary Fig. 5c** shows that alanine substitution mutants of the interface residues PapC W767 and F732 are defective for pilus assembly.

Point 4: The authors state that the NTD “pushes” the newly recruited chaperone-subunit towards the CTDs, while it is not shown if this is simply synchronized movement driven by some other factor, or if the NTD indeed exerts a “pushing” force.

Response: We agree that we do not have evidence for an active role of the NTD in pushing the bound subunit toward the CTDs. Therefore, we have changed the sentence on lines 191-194 as follows: “...which suggests that the usher NTD conveys the newly recruited chaperone-subunit toward the CTDs...” We similarly changed the sentence on lines 273-275 as follows: “...conveying the bound chaperone-subunit complex toward the CTDs side of the usher...”

Point 5: The authors claim there is “upward movement” when comparing the location of PapG in Conformers II and III (Fig 3b), but the structures presented seem to show no difference in the location of PapG relative to the PapC usher.

Response: We have revised **Fig. 3b** to indicate the 2.1 Å upward movement of PapG at the top from Conformers II to III, along with the 21 and 28 Å lateral movements of NTD and CTD2, respectively.

Reviewer #2

Minor points

Point 1: Lines 32 and 88: I would suggest that molecular mechanisms of pilus assembly and function could rather facilitate development of “pilus assembly-targeted” rather than “pilus-targeted” therapeutics. Structures of the pilus fibres may lead to “pilus-targeted” therapeutics.

Response: We changed lines 29-30 as follows: “Such knowledge may facilitate the development of therapeutics that disrupt pilus biogenesis as an alternative...”. We also changed lines 86-88 as follows: “...and allow for rational design of therapeutics that disrupt pilus biogenesis.”

Point 2: Lines 36/37: Perhaps you could insert the following reference: M. D. Crespo et al., “Quality control of disulfide bond formation in pilus subunits by the chaperone FimC,” *Nat. Chem. Biol.*, vol. 8, no. 8, pp. 707–713, Aug. 2012.

Response: The reference has been added.

Point 3: Lines 70-71: For completion, could include references to pilus fibre structures?

Response: We have added the following references: Hospenthal *et al.*, *Cell* **164**, 269-278 (2016); and Hospenthal *et al.*, *Structure* **25**, 1829-1838 (2017).

Point 4: Line 145: Consider rephrasing “dynamics”.

Response: We have revised the sentence on lines 157-158 as follows: “In contrast, Conformers I and II capture new states of the usher during pilus assembly.”

Point 5: Line 160: Perhaps consider removing “observed” and simply say “as in Conformer I”, to avoid any confusion about the fact that no CTD2 density is visible in this state.

Response: We have deleted “observed” from the sentence on line 173 of the revised manuscript.

Point 6: Line 173: Are all figure panels referred to here relevant for this statement?

Response: All figures cited are relevant, and we neglected to cite **Supplementary Fig. 5**, which is also relevant. To simplify the text and make this correction, we have changed the figure citation (lines 186-187) to: “(**Figs. 1-3, Supplementary Fig 5, Supplementary Movie 1**).”

Point 7: Line 223: This statement could “suggest” that all of these residues are conserved, which is not entirely true. Slight rephrasing? What is “O” in the alignment shown in Extended Data Fig. 6?

Response: Although only Y32 has high identity among the ushers, all of the residues chosen for mutagenesis are located in regions of high conservation. Other highly conserved residues located at these interfaces have previously been analyzed, as noted in the manuscript (see paragraph starting on line 218), supporting a critical role for these sites of usher-chaperone subunit interactions. We have changed the text of the revised manuscript (lines 236-239) as follows: “The residues newly revealed by our structures to be involved in contacts between the PapC periplasmic domains and PapD chaperone are located in regions that are highly conserved among members of the usher superfamily (**Supplementary Fig. 6**).”

The “O”s in the alignments are in fact “Q”s in which the bottoms of the Qs were cut off by the alignment spacing. We have corrected the line spacing in the revised **Supplementary Fig. 6** so that the full “Q”s are visible.

Point 8: Line 244: By listing all interacting residues in the bracket, it is unclear which belong to the plug and which to the NTD.

Response: We have clarified the text and added a reference to **Supplementary Fig. 9**, which shows the plug-NTD interaction, as follows (lines 258-260): “A plug-NTD interface is formed by interactions involving hydrophobic and charged residues (plug residues P254, R256, D297; NTD residues Y128, D130, W133) (**Supplementary Fig. 9**).”

Point 9: Lines 252-253: The observation that there is no direct path for the plug from the “closed” to the “open” state, does not really suggest that the “closed” state structure alone requires further validation.

Response: We have removed the second part of the sentence. The revised sentence (lines 267-268) now reads: “Additional studies will be needed to determine how this transition occurs.”

Point 10: Line 353/Extended Data Table 1: The total electron dose in the table is 50 e-/Å², the text suggests 60 e-/Å².

Response: The dose rate in the table has been changed to 60 e-/Å².

Point 11: Line 354: Rather than stating how long each frame was exposed for, perhaps state the dose/frame.

Response: The sentence (line 370) has been changed to read: "The total dose was divided into a 30-frame movie with a dose per frame of 2 e-/Å²."

Point 12: Line 361/Extended Data Table 1: Typo? Relion-3.0?

Response: Line 378 has been corrected to Relion-3.0.

Point 13: Lines 374-376: Question. Did you try to individually refine these three 3D classes? Or class 1 and 3 combined? Are these three combined classes really similar enough and could this explain the modest resolution of Conformer I?

Response: We followed the reviewer's suggestion and refined these classes individually, leading to final maps at 7.7 Å (cls1), 8 Å (cls2), and 7.4 Å (cls3) resolution. As shown in the figure above, we found that the three maps are superimposable with the Conformer I map

(combined from all three classes). We did notice that the PapD chaperone and the NTD and CTD of the PapC usher shift slightly. However, the movements are too small to assign them as new conformations.

Figures

Point 1: Figure 1c: Could you please label CTD1 in conformer III?

Response: We added the label as suggested in the revised **Fig. 1c**.

Point 2: Extended Data Fig. 5: I46 mentioned in line 216 is not indicated on the figure. What do the red asterisks indicate? Residues chosen for mutation? Mention in legend.

Response: We labeled residue I46 in the revised **Supplementary 5b**. The red asterisks indicate residues that were mutated and tested by the HA assay shown in panel **c**. We changed the text in the figure legend as follows: "**c**, Hemagglutination assay results comparing the ability of wild-type (WT) PapC and PapC point mutants (denoted by red asterisks in panels **a** and **b**) to assemble adhesive P pili."

Point 3: Extended Data Fig. 6: What is "O"

Response: The "O"s in the figure are in fact "Q"s in which the bottoms of the Qs were cut off by the alignment spacing. We have corrected the line spacing in the revised **Supplementary Fig. 6** so that the full "Q"s are visible.

Reviewer #3

Point 1: Extended data figure 4d: It seems that there is no helical fold, but only sheets.

Response: We corrected the error in the revised figure legend.

Point 2: Extended data figure 5b: because of the low resolution of conformer III, it would be more convincing to show the density map of the residues involved in the interface between PapC-NTD and -CTD2.

Response: We have revised **Supplementary Fig. 5a-b** to include superimposed 3D density in the zoomed windows showing the specific interactions.

Point 3: Extended data figure 5c: regarding the point mutations, a lower HA titer could result from less stable, rather than less functional usher. For instance, western-blot of PapC mutant would help to show normal expression and addressing to the outer-membrane.

Response: For each of the PapC mutants constructed, we confirmed proper expression and folding in the bacterial outer membrane, by immunoblotting and using a heat-modifiable mobility assay. This is noted in the Methods section of the manuscript (lines 330-332). In the revised manuscript, we added blots showing these results as a new panel **d** in **Supplementary Fig. 5**. The figure legend has been revised as follows: "**d**, Expression levels and folding of the PapC

mutants analyzed in panel **c**. OM fractions from bacteria expressing WT or mutated PapC were incubated at 25°C or 95°C in SDS-PAGE sample buffer, subjected to SDS-PAGE, and blotted with anti-His-tag antibodies. The presence of the faster-migrating folded monomer band in the 25°C-treated samples indicates proper folding of the usher in the OM.”

Point 4: Manuscript line 222-224: except Y32, most of the residues of the interacting interface do not seem to be conserved according to extended data figure 6.

Response: Although only Y32 has high identity among the ushers, all of the residues chosen for mutagenesis are located in regions of high conservation. Other highly conserved residues located at these interfaces have previously been analyzed, as noted in the manuscript (see paragraph starting on line 218), supporting a critical role for these sites of usher-chaperone subunit interactions. We have changed the text of the revised manuscript (lines 236-239) as follows: “The residues newly revealed by our structures to be involved in contacts between the PapC periplasmic domains and PapD chaperone are located in regions that are highly conserved among members of the usher superfamily (**Supplementary Fig. 6**).”

Point 5: Extended data figure 6: it would be interesting to add bacterial species of the different usher proteins.

Response: The names of the bacterial species are too long to integrate into the figure. In the revised manuscript, we have added the bacterial names to figure legend. PapC, FimD, MrkC, AfaC belong to *E. coli*, SafC and PefC belong to *Salmonella enterica*, FaeD belongs to *Erwinia amylovora*, and PsaC belongs to *Yersinia pestis*.

REVIEWERS' COMMENTS

Reviewer #1 (Remarks to the Author):

The authors have successfully addressed the previously-described concerns, and have made the appropriate corrections. The authors have described how the relatively low resolution of 7-8Å was not particularly problematic for domain assignment in the P pilus system, as the Ig-like folds that subunits adopt are relatively rigid and that the Conformer II, solved at higher resolution, served as an appropriate template for domain placement. The authors also explained how the modular and iterative process of pilus assembly allows the assembly mechanism to be elucidated by their structure, despite the absence of PapE and PapF. All other minor concerns were successfully addressed. This is an outstanding manuscript.

Reviewer #2 (Remarks to the Author):

I am satisfied that the authors have addressed all my comments.

Reviewer #3 (Remarks to the Author):

I am happy with the responses of the authors. I would recommend publication of the revised manuscript.